# Comparative Structural and Compositional Analyses of Cow, Buffalo, Goat and Sheep Cream

**DOI:** 10.3390/foods10112643

**Published:** 2021-11-01

**Authors:** Valeria D. Felice, Rebecca A. Owens, Deirdre Kennedy, Sean A. Hogan, Jonathan A. Lane

**Affiliations:** 1H&H Group, Global Research and Technology Centre, P61 C996 Cork, Ireland; valeria.felice@hh.global; 2Department of Biology, Maynooth University, W23 F2H6 Maynooth, Ireland; Rebecca.Owens@mu.ie; 3Teagasc Food Research Centre, Moorepark, Fermoy, Co., P61 C996 Cork, Ireland; deirdre.kennedy@teagasc.ie (D.K.); Sean.Hogan@teagasc.ie (S.A.H.)

**Keywords:** cow, goat, sheep, buffalo, cream, milk fat globules

## Abstract

Factors affecting milk and milk fraction composition, such as cream, are poorly understood, with most research and human health application associated with cow cream. In this study, proteomic and lipidomic analyses were performed on cow, goat, sheep and *Bubalus bubalis* (from now on referred to as buffalo), bulk milk cream samples. Confocal laser scanning microscopy was used to determine the composition, including protein, lipid and their glycoconjugates, and the structure of the milk fat globules. BLAST2GO was used to annotate functional indicators of cream protein. Functional annotation of protein highlighted a broad level of similarity between species. However, investigation of specific biological process terms revealed distinct differences in antigen processing and presentation, activation, and production of molecular mediators of the immune response. Lipid analyses revealed that saturated fatty acids were lowest in sheep cream and similar in the cream of the other species. Palmitic acid was highest in cow and lowest in sheep cream. Cow and sheep milk fat globules were associated with thick patches of protein on the surface, while buffalo and goat milk fat globules were associated with larger areas of aggregated protein and significant surface adsorbed protein, respectively. This study highlights the differences between cow, goat, sheep, and buffalo milk cream, which can be used to support their potential application in functional foods such as infant milk formula.

## 1. Introduction

The use of domestic animal milk and milk fractions for human nutrition for all life stages is well established [1]. Although significant technological advances associated with the characterisation of milk composition have been achieved, there remains limited detailed information on the composition, and change in composition over lactation, of milk and milk fractions from certain mammals. Furthermore, factors influencing milk and milk fraction composition are yet to be understood. As reported by the Food and Agriculture Organization of the United Nations (FAOSTAT data, 2016), milk is produced and consumed in all the world’s countries with milk production predominantly associated with cow (82.7%), buffalo (13.3%), goat (2.3%), sheep (1.3%) and camel (0.4%). Milk and dairy products account for nearly 14% of the global agricultural trade, with a preference for whole milk powder and skimmed milk powder. 

The most investigated milk source in terms of production, composition and associated function is cow milk. A factor driving these investigations is the common use of cow milk for infant milk formula. Most recently, O’Callaghan and colleagues [2,3], have demonstrated the impact of feeding systems, predominantly indoor feeding using total mixed-ration diets and pasture-feeding, on milk composition, processing, and end-product. Indeed, research has clearly demonstrated that feeding systems influence milk fat, protein, mineral, vitamin, amino acid, and fatty acid composition [4]. Furthermore, research is now accumulating on the impact of the feeding system on the taste of milk and milk-based products such as butter. For example, pasture-feeding produces milk perceived to have a grassy, cowy, mothball and/or barny flavour, which corresponds with the presence of higher concentrations of *p*-cresol in the headspace of pasture milk [5]. Cow milk, its fractions and isolated milk components have been associated with numerous health benefits. Whey proteins such as osteopontin, lactoferrin, α-lactalbumin and immunoglobulin G have been linked with anti-microbial, anti-viral, immunomodulatory, metabolic and anti-carcinogenic properties [6]. 

Although buffalo, goat and sheep milk have been an important part of human nutrition for millennia, detailed knowledge of the composition and function of these types of milk is limited when compared with cow milk. Goat milk and milk fractions have been studied for their digestive properties. Interestingly, notable differences have been reported for cow and goat milk [7,8,9]. Indeed, Hodgkinson and colleagues have reported different digestive behaviours in an *in vitro* digestion model between goat and cow milk with goat casein tending to be more efficiently digested, when compared to cow casein, with overall peptides profiles from goat and cow milk, post digestion, being distinctly different [8,9]. It is important to note that others have observed both similar and conflicting results [10,11]. Thus, further studies in this area appear to be warranted. Sheep milk has been predominantly used to produce cheese and yogurt; however, increased production and availability of sheep milk has positioned it as a potential source of health-promoting milk fractions. As reported by Claeys and colleagues [12], sheep milk is an abundant source of protein with molecular forms and amino acid compositions that support its digestibility and nutritional value. Functional research on sheep milk remains limited. Similarly, to sheep milk, buffalo milk research has mainly focused on its safety, production, and processing with a core focus on the quality of end products such as cheese and yogurt. Hence, there is a need to further understand these milk and milk fractions to fully unlock their potential as sources of health-promoting ingredients. 

In this study, we aimed to perform comparative analyses of the structure, composition, and potential function of cow, goat, sheep and buffalo cream. Our focus on the milk fat fraction was associated with the fact that the cow Milk Fat Globule Membrane (MFGM), enriched in cow cream, has gained industrial interest due to pre-clinical and clinical evidence associated with the development of the infant immune and gastrointestinal system, cognitive function, protection from infection, and cardiovascular and muscular health [13]. Thus, we aim to understand if cow, buffalo, goat and sheep cream sources could have similar structural, compositional, and therefore, functional profiles and to identify their potential for inclusion in functional foods. 

## 2. Materials and Methods

### 2.1. Sample Collection

All domestic animal milks were collected from bulk tanks from commercial dairy farms under controlled conditions. Cow, sheep, goat and buffalo milks were sourced from Isigny Sainte Mere (France) (Breeds, Holstein > Normandy > Jersey > Others), Beechmount Farm—Sheep Milk Ireland (Ireland) (Breeds, East Friesian and Lacaune), Ardsallagh (Ireland) (Breed, Saanen) and Macroom Bubalus bubalis Cheese (Ireland) (Breed, Mediterranean Italian), respectively. Cream was separated from raw, unpasteurised milk through a one-step centrifugation process using a disc-bowl centrifuge (Armfield Ltd., Ringwood, England) (37 °C, 6000 rpm), except for cow milk which was separated industrially. Separated samples were stored and shipped at 4 °C and analysed within 48 h except for cream samples used for proteomic analysis, which were lyophilized and stored at −20 °C prior to analysis. 

### 2.2. Cream Composition Analysis

#### 2.2.1. Total Protein Content

Total protein quantification was determined using the Kjeldahl method and a nitrogen-to-milk protein conversion factor of 6.38. This method was based on the International standard ISO 8968-3:2004/IDF 20-3:2004.

#### 2.2.2. Phospholipid Profiling 

Quantitative 31P-NMR spectroscopy was performed on a Bruker Avance III 600 MHz with an automatic sample changer and BBO cryoprobe at Spectral Service (Cologne, Germany). 

#### 2.2.3. Fatty Acid Profiling 

The extraction of crude fat from cream for full fatty acid characterisation and determination of the composition of fatty acids in the 2-position of the triglyceride molecules was performed by ITERG (Canéjan, France).

### 2.3. Proteomics

#### 2.3.1. Protein Preparation

Protein samples were prepared by dissolving cream powder in SDS buffer (4% (*w/v*) SDS, 0.1 M Tris-HCl pH 7.4) at a final concentration of 100 mg/mL and incubated at 95 °C, 10 min. Samples were centrifuged at 14,000 rcf for 15 min and the soluble protein fraction was isolated. Samples in SDS buffer were mixed (4:1) with 5X sample buffer (50% (*v/v*) glycerol, 1% (*w/v*) SDS, 30 mM Tris-HCl, 2.5% (*v/v*) β-mercaptoethanol, 0.006% (*w/v*) bromophenol blue), incubated at 95 °C for 5 min and loaded onto 12% acrylamide/methylene bisacrylamide gels [14] for SDS-polyacrylamide gel electrophoresis (SDS-PAGE). Electrophoresis proceeded until the samples had entered the top 1 cm of the resolving gel (for protein digestion), or on separate gels until the dye front had reached the end of the resolving gel (for visualisation of protein bands). Gels were stained with Coomassie Blue and de-stained to remove background. Condensed protein bands were excised from the top 1 cm of the resolving gel and in-gel digestion was performed according to Shevchenko et al. [15]. Sample clean-up was performed using Zip Tips with C18 resin (Millipore) and dried peptide mixtures were stored at −70 °C.

#### 2.3.2. Sample Clean-Up and Q-Exactive Analysis

Peptide extracts were re-suspended in 0.05% (*v/v*) trifluoroacetic acid, 2% acetonitrile and were analysed by LC-MS/MS using the Thermo Q-Exactive mass spectrometer coupled to a Dionex Ultimate 3000 RSLCnano [16]. Peptides were separated in-line on an EasySpray PepMap C18 column (500 mm × 75 µm with 2 µm particles) on a 14 to 35% B gradient as outlined in Morrin et al. [17]. The data-dependent acquisition was used with a Top15 method for MS/MS scans. MaxQuant (v 1.6.2.10) was used for protein identification [18,19]. Raw files from cream samples were searched against a combined protein database consisting of the *Bos taurus* (cow), *Ovis aries* (sheep), *Capra hircus* (goat) (all downloaded from Uniprot, 05 September 2019) or *Bubalus bubalis* (water buffalo) database (downloaded from NCBI 06 November 2019). A modified contaminants database was also used, excluding *B. taurus* entries. The false discovery rate (FDR) was set to 1% for both peptides and proteins. Data organisation and analysis were performed using Perseus (v 1.6.6.0) [19]. Protein groups were filtered to exclude those matching a decoy or protein contaminants database, or those identified by only a single peptide.

### 2.4. Confocal Laser Scanning Microscopy

The confocal methods carried out, were based on [20,21] with minor modifications. Confocal laser scanning microscopy (CLSM) analysis of MFGs was performed using a confocal laser scanning microscope (LeicaSP5, Leica Microsystems CMS GmbH, Wetzlar, Germany). Experiments used an argon laser with a 488 nm excitation wavelength, a diode laser with a 561 nm excitation and a helium neon red laser line (excitation 633 nm). 

#### 2.4.1. Protein/Fat Labelling

Neutral lipids were stained with Nile Red 0.10 g/L in polyethylene glycol 200 (Sigma Aldrich, Wicklow, Ireland) with emission collected in the range 500–530 nm. Protein was labelled using Fast Green (0.01 g/L in water) fluorescent dye (Sigma Aldrich) with emission collected between 650–700 nm. 50 µL of a 3:1 mixture of Nile Red: Fast Green was added to 1 mL of cream/solution, which was then vortexed and 20 µL deposited on a glass slide before imaging. A 63x oil immersion objective was used to acquire images taken at 1024 × 1024 pixels.

#### 2.4.2. Phospholipid and Carbohydrate Labelling

The fluorescent dye 1,2-dioleoyl-sn-glycero-3-phosphoethanolamine-N-(lissamine rhodamine B sulfonyl) (Rh-PE, Avanti Polar lipids Inc., Birmingham, UK) was made up to 1 mg/mL in chloroform and used to label the phospholipids by adding 50 µL of the solution to 1 mL of cream. Emission was collected in the range 570–625 nm.

Wheat Germ Agglutinin Alexa Fluor 488 (WGA, Cergy Pontoise, France) was made up to 1 mg/mL in low salt TBS (20 mM Tris-HCl, 100 mM NaCl, 1 mM CaCl2, 1 mM MgCl_2_, pH 7.2) and used to label and locate carbohydrate moieties. Fifty µL of solution was added to 1 mL of cream. Emission was collected in the range 490–550 nm. For samples single and dual stained with Rh-PE and WGA, samples were kept at room temperature in the dark for a minimum of 6 h before analysis. 0.5% (*w/w*) low melting point agarose (held at 45 °C until required) (Thermofisher Scientific, Waltham, MA USA) was added to all single and dual stained samples before imaging 10 µL of sample and 20 µL of the agarose were deposited onto slides and mixed gently before a coverslip was added. A 63x oil immersion objective was used to acquire images taken at 1024 × 1024 pixels.

## 3. Results and Discussion

### 3.1. Total Protein Content in Cow, Buffalo, Goat and Sheep Cream Samples

The total protein content for domestic animal milks is well established. For example, it has been reported that the natural total protein content in mature cow, goat, sheep and buffalo milk is 3%, 2.75%, 6.36% and 4.27%, respectively [22,23,24,25]. The total protein content recorded in milk and, indeed, the cream is influenced by numerous factors, including animal species, lactation period, breed, feed, and laboratory methodology and practise [24,25]. In this study, the total protein content of the cream samples analysed, was determined as 5.47, 6.03, 3.68 and 5.12% for cow, sheep, buffalo and goat, respectively (*n* = 1; Table 1). To our knowledge, this is the first study investigating the protein content in the cream of goat, sheep and buffalo milks.

### 3.2. Proteomic Analyses of Cream Samples from Four Species

Proteins isolated from cow, sheep, goat and buffalo cream were analysed by SDS-PAGE and revealed some distinct protein banding patterns in the 40–70 kDa range (*n* = 1; Figure 1). SDS-PAGE-based proteomic profiling is limited by the dynamic range of the proteins in cream, with mainly highly abundant proteins visible using this method. To identify the proteins, qualitative analysis was performed using shotgun proteomics. Concatenated databases were prepared composed of the proteomes of *Bos taurus*, *Capra hircus* and *Ovis aries* (downloaded from Uniprot 05 September 2019), and *Bubalus bubalis* (water buffalo) database (downloaded from NCBI 06 November 2019). Combined analysis of proteins from buffalo, cow, goat and sheep cream was conducted using the four species database. In the 4 species analysis, 1230 protein groups were identified in total, with peptides from all species matching to 323 shared protein groups (*n* = 1; Figure 2). Qualitative proteomic analysis identified 669, 802, 753 and 685 protein groups from cow, sheep, goat and buffalo cream, respectively. Several protein groups were exclusively matched by peptides from buffalo (78 protein groups), cow (106 protein groups), sheep (128 protein groups), or goat (92 proteins groups) cream samples (*n* = 1; Figure 2). Buffalo and cow samples shared 89 protein groups, which were not matched to goat or sheep samples, while 133 protein groups were exclusively shared by goat and sheep. 

### 3.3. Functional Analysis of Cream Samples by Species

Functional analysis of all protein groups detected across the different cream samples was conducted using BLAST2GO analysis against a local BLAST database generated from the Bovidae entries in Uniprot (downloaded 24 May 2020) and gene ontology (GO) term mapping and annotation. GO terms are divided into 3 categories: biological process (BP), molecular function (MF) and cellular component (CC). For all species, “biological process” was the category with annotations for the highest number of protein groups. A summary of BP associated terms along with the number of protein groups identified from the cream samples of the associated species is shown in Figure 3 (*n* = 1) and includes: cellular process, biological regulation, metabolic process, regulation of biological process, response to stimulus, localization, multicellular organismal process, positive regulation of biological process, signalling, negative regulation of the biological process, developmental process, immune system process, interspecies interaction between organisms. An additional representation of this data is included alongside (right) showing the protein groups normalised to the total number of functionally annotated protein groups from that species. In addition to biological processes, GO molecular function (*n* = 1; Appendix A) and cellular component (*n* = 1; Appendix A) are shown for the four species comparisons, with absolute numbers of detected proteins (left) and the data normalised to the number of proteins with associated GO category terms (right) for each figure. There were higher numbers of detected proteins associated with developmental process, multicellular organismal process, response to stimulus and regulation of the biological process in the goat and sheep samples compared to buffalo and cow samples. Following normalisation, most terms showed similar levels across each of the species analysed. However, these data were limited to high-level broad BP terms; thus, to distinguish more subtle differences, further investigation of more specific BP terms and individual protein groups was undertaken.

Protein components of cream like those within the milk far globule membrane (MFGM) have been associated with several health benefits, including and predominantly immune function [26]. Hence, we have further focused our analysis on immune system-related terms. A summary of immune system-related terms is shown in Figure 4 (*n* = 1) and includes regulation of immune system process, immune effector process, activation of immune response, immune system development, antigen processing and presentation, leukocyte migration, and myeloid cell homeostasis. A certain level of variability was observed in the GO terms among the species. Among those, we focused on antigen processing and presentation, activation of immune response and production of molecular mediator of immune response. 

The antigen process and presentation was associated with 21 protein groups for cow, 23 for buffalo, 19 for goat and 23 for sheep cream (*n* = 1; Figure 4). When the proteins associated with this GO term were analysed, we observed several members of the RAS family (*n* = 15; Table 2). Of these proteins, eight were present in all species; two were present in cow, goat and sheep but not in buffalo; two were present in buffalo and goat and three proteins were specific for either buffalo, goat or sheep. Members of the RAS family have also been previously detected in human skim milk [27] as well as in the MFGM fraction of buffalo milk [28]. Several Ig-like domain-containing proteins were also detected across all species (Table 2). The proteasome activator complex subunit 1 was only detected in buffalo and sheep while the protein prosaposin was only detected in cow and sheep (Table 2). Prosaposin is the precursor protein for four lysosomal activator proteins, saposins A-D, which act as sphingolipid activator proteins that facilitate the hydrolysis of sphingolipids via lysosomal hydrolases [29]. In addition to this function, the full-length protein can also be secreted into several secretory fluids, including milk [30], where it acts as neurotrophic factor, promoting cell survival, neurite outgrowth and differentiation in a cholinergic cell line [31,32]. Prosaposin has been shown to be present in human milk as well as in milk of other species including cow and goat [30,33]. However, in our study, this protein was only detected in cow and sheep cream but not in buffalo or goat cream. This could either indicate that it is present at concentrations below the limit of detection or that levels could differ in cream versus milk. 

Within the GO term activation of the immune response, 35 proteins were detected for cow, 36 for buffalo, 41 for goat, and 43 for sheep (*n* = 1; Figure 4). From this group, several members of the complement family were detected across all species, with few uniquely detected in some species (Table 3). The complement system includes plasma proteins that coat extracellular pathogens, facilitating their removal by phagocytes or direct killing. MFGM-related proteins associated with activation of the immune response, such as butyrophilin subfamily 1 member A1 and tyrosine-protein kinase, were also identified across all species (Table 3). An interesting protein that was observed in all species except cow is pentraxin (Table 3). This protein family includes the short pentraxins serum amyloid P component, C-reactive protein, and the prototypic long pentraxin PTX3. Pentraxins are part of the pattern recognition receptors (PRRs), which are specialized in recognition of highly conserved motifs expressed by microbes. These receptors are classified based on their localization and include (1) endocytic PRRs, (2) signaling PRRs and (3) soluble PRRs, which include pentraxins, collectins and ficolins. These proteins can bind selected microbes and facilitate their disposal by phagocytes [34]. PTX3 has been shown to be present in human breast milk, potentially contributing to the protection of infants against infections [35]. Interestingly, Mudaliar and colleagues have shown upregulation of PTX3 in an experimental model of cow mastitis [36]. This protein is indeed produced at sites of infection and inflammation by both somatic and immune cells, and its glycosylation has been implicated in modulating protein functions, including the modulation of the complement system through the interaction with the complement component C1q (reviewed by [37]). Interestingly, our analysis has identified complement C1q A chain and B chain in goat and sheep, and in sheep, respectively, while neither were detected in cow and buffalo (Table 3). This could again either indicate that these proteins are present at concentrations below the limit of detection. 

Another protein that was detected in all species, except cow, is the major prion protein (PrP) (Table 3). This protein has received considerable attention due to its role in the pathogenesis of prion disease or spongiform encephalopathies (TSEs), affecting both humans and animals. The normal PrP (PrP^C^) is apparently benign. However, it is capable of post-translational misfolding into an abnormal and infectious isoform (PrP^Sc^) [38]. The normal PrP^c^ has been detected, with differences in expression, in the mammary gland of domestic ruminants such as cow, sheep and goat [39,40,41], as well as in humans, cow, sheep and goat milk [42,43]. In contrast to these findings, Didier and colleagues [40], did not detect PrP^C^ in any of the cow milk fractions analysed in their study, including cream, which is consistent with what we have observed in our study. However, this is probably due to the methodology used in Franscini’s study, which leads to a higher concentration of the prion content. Interestingly, PrP^C^ was easily detected in sheep and goat milk fractions, with the highest levels observed in the cream fraction [40]. This evidence suggests that PrP may be present in our cow cream sample but below the limits of detection, while the higher levels present in goat and sheep cream allowed for its detection. No evidence has been shown to date for the presence of PrP in buffalo milk. 

Alpha-2-macrogobulin (A2M) is a plasma protein involved in the inhibition of a wide range of serum proteases. While we have only detected this protein in goat and sheep cream (Table 3), in a previous study, A2M was detected in cow milk, with the highest concentration in the first milking [44]. This could be related to the capacity of A2M to inhibit the protease, hence preventing the degradation of biologically active proteins (i.e., immunoglobulins) in the intestine of the newborn. Furthermore, A2M plays a key role as a humoral defence barrier against pathogens, binding host or foreign peptides and particles [45]. Interestingly, higher concentrations of A2M have been detected in mastitic cow’s milk compared to normal milk and this was related to the degree of mastitis [44]. 

Toll-like receptor 4 (TLR4) was detected only in buffalo under the GO terms activation of immune response and production of molecular mediator of the immune system (Table 3 and Table 4). As with pentraxins, Toll-like receptors are PRRs. TLR4 recognises bacterial lipopolysaccharide (LPS), which leads to the activation of the intracellular signalling pathway, NF-κB and the subsequent production of inflammatory cytokine, activating the innate immune system [46]. The presence of TLR4 has been previously investigated in breast milk (MFGM and skimmed fraction) by Cattaneo and colleagues [47]. In this study, TLR4 was not detected in the samples analysed; however, the group concluded that this result cannot assure the total absence of this receptor in milk but could also be due to the limit of detection of the instrument or the methodology used. Interestingly, Cao and colleagues [48], detected this receptor in both colostrum and mature milk MFGM with higher levels found in the latter. TLR4 was also detected in the sheep milk whey protein fraction [49], while to our knowledge, there is no evidence of the presence of TLR4 in milk or milk fractions in cow, goat, and buffalo. 

Under the GO term production of molecular mediator of the immune system, 26 proteins were detected for cow, 19 for buffalo, 24 for goat and 21 for sheep (*n* = 1; Figure 4). MFGM related proteins such as platelet glycoprotein 4/CD36, Toll-like receptor 2, apolipoprotein A1 have been identified across all species as well as several Ig-like domain-containing proteins (Table 4). 

An interesting protein that was detected in cow, goat and sheep cream but not in buffalo is the transforming growth factor beta-2 proprotein (Table 4). The protein intensities, however, were low in these three species; thus, it could be the case that even lower levels were present in buffalo that went undetected as opposed to this protein being absent. The transforming growth factor beta-2 proprotein is the precursor of the latency-associated peptide (LAP) and transforming growth factor beta-2 (TGF-β2) chains, which constitute the regulatory and active subunit of TGF-β2, respectively. TGF-β, is the most abundant cytokine in breast milk, and includes TGF-β1 and TGF-β2, with the latter being predominant [50,51]. Breast milk TGF-β has gained increasing interest as it is involved in maintaining intestinal homeostasis, regulating inflammatory responses and promoting the development of neonatal oral tolerance [52,53]. Breast milk TGFβ2 has also been shown to be associated with the neonatal gut microbial composition and increased richness, evenness, and diversity [54]. The relationship between TGFβ and allergies prevention is controversial. An initial systematic review showed that high concentrations of TGF-β1 or TGF-β2 in human milk, were positively associated with a reduction in immunological outcomes of allergies in seven of 12 studies included [55]. However, the result of a recent systemic review was in contrast with this conclusion, not finding strong evidence of associations between any isoform of human milk TGF-β and allergic outcomes [56]. Both TGF-β isoforms have also been identified in cow milk with TGF-β2 being again the most abundant [57]. In cow milk both TGF-β2 and TGF-β1 levels has been shown to increase during mastitis induced by *E. coli* [58]. To our knowledge this is the first time that TGF-β2 has been shown in goat and sheep cream.

Another cytokine that was detected in cow, goat and sheep but not in buffalo is the macrophage migration inhibitory factor (MIF) (Table 4). This cytokine is released in response to proinflammatory stimuli and inhibits the migration of macrophages, enhancing their phagocytic activity. The presence of MIF in human milk was identified for the first time by Magi and colleagues [59], in the milk aqueous phase and inside milk fat globules. This was further confirmed by Vigh and colleagues [60], who detected a high concentration of MIF in breast milk, especially during the first month of lactation. No studies to date have shown the presence of MIF in the milk of any of the species subject of this study.

### 3.4. Lipid Analysis of Cream Samples from Four Species

Milk fat contains several thousand lipid species and is the most complex material in nature in terms of lipid composition. The fatty acid (FA) composition of milk fat triglycerides (which account for approximately 98% of total milk lipids) are affected by several factors, including species, breed, diet, and seasonality [61]. The milk fat of domesticated animals is composed primarily of two major fractions: long-chain (50–70%) and short-chain FAs (30–50%). 

Long-chain FAs (C18-24) are typically derived from the diet, whereas short-chain FAs (C4 to C14 and some C16), are synthesized de novo by the mammary gland. Saturated FAs in ruminant milk accounts for approximately 60 to 70% of the total. 

In the present study, total fat (*n* = 1; Table 5) and fatty acid content of all cream samples were determined. The levels of saturated fatty acids (SFA) were lowest in sheep cream and broadly similar in the cream of the other species examined (*n* = 1; Table 6). MacGibbon and Taylor [62] and Markiewicz-Kęszycka et al. [63] also reported lower total SFA in sheep milk compared to cow and goat. In terms of short-chain FAs, buffalo cream contained the highest levels of butyric acid (C4:0), whereas the levels of caprylic (C8:0) and capric (C10:0) acids were much higher in sheep and goat milk fat than that of cow and buffalo. Caproic, caprylic and capric acids are so termed because of their high proportions in goat milk. Variations in de novo synthesis of short-chain FAs are controlled by several genes expressed in the mammary gland [64], and may account for the differences observed between species. Palmitic acid (C16:0) was highest in cow cream (32.2%) and was, with the exception of sheep milk fat (19.7% *w/w*), the most abundant of all FAs. Castro-Gomez et al. [65] reported greater similarities in C16:0 between cow, sheep and goat milks with values of 32, 29 and 28%, respectively. Palmitic acid, along with a number of other high-melting-point FAs, is the main contributor to hardness in milk fat products as it remains solid at room temperature.

Oleic acid (C18:1), the most abundant monounsaturated FA (MUFA) in mammalian milk fat, was highest in sheep milk and was very similar in the other three species. In contrast with C16:0, it is a low-melting FA and thus, contributes to the liquid phase in semi-crystalline dairy products, thereby contributing to softness and spreadability. Other MUFAs in ruminant milk include myristoleic acid (C14:1), which was much lower in sheep and goat cream compared to that of cow or bubalus bubalis; palmitoleic (C16:1), lowest in goat milk, and vaccenic acid (C18:1 *trans*), a naturally occurring *trans* FA with cholesterol-lowering functionality.

Generally speaking, milk fat of non-ruminants has a higher level of polyunsaturated FAs (PUFA) than that of ruminants, due predominantly to direct absorption from the diet [62]. Sheep milk fat had higher levels of PUFA than the other species examined, a finding supported by previous reports [62,66,67]. This difference is attributable, in the main, to the high levels of linoleic acids (C18:2), a diverse group of FA isomers, which includes α-linoleic acid (C18:2 *n*-6, cis), one of two essential FAs, and a number of conjugated linoleic acid isomers (CLA), a group of FAs associated with a range of health benefits. Serra et al. [67] reported that feeding with linseed oil significantly increased levels of linolenic (C18:3) and rumenic acid (C18:2 CLA) in sheep milk, with both of these also present in higher quantities at the *sn-2* (middle) position of the triglyceride molecule. Markiewicz-Kęszycka et al. [63] reported that sheep and goat milks are usually richer in CLA than cow milk and that the concentration of CLA in the milk fat of sheep milk is greater than that in goat milk, with this effect due to differences in the mRNA of their mammary adipocytes. The level of C18:3, the other fatty acid essential to human nutritional needs, was significantly lower in goat cream, compared to the other species. Arachidonic acid (C20:4, *n*-6) was approximately twice as high in sheep cream as cow, goat and buffalo milk fat. 

In general, the FA profiles of cow and buffalo milk fat were, with the exception of butyric, palmitic and linoleic acids, similar in composition. Pegolo et al. [68] reported that *buffalo* and cow species have comparable average milk FA although others have reported greater variations [69,70]. The lipid composition of milk responds to changes in diet in a more pronounced way compared to other macro-constituents such as proteins, which are determined largely by genetics. FA profile can be manipulated through direct addition (supplementation) of FAs in the diet, alteration of rumen conditions, or via biotransformation, in the rumen, of dietary FAs to other (long-chain) FA species. Pasture feeding increases the concentration of certain milk FAs, mainly C18:0 C18:1 C18:3 and CLA, and decreases saturated FAs from C10:0 to C16:0 [46,54]. Mixed-ration (concentrated) feeds have been reported to yield higher levels of C16:0 and lower levels of nutritionally beneficial FAs [5]. The health implications of milk fat consumption have, for many years, generated much controversy and confusion, resulting in commonly held negative perceptions. More recently, meta-analytical studies have redressed this imbalance and have contributed to a gradual shift in scientific opinion that considers the contribution of SFAs to human health to be less detrimental than previously thought. This topic has been the subject of recent reviews by Mohan et al. [61] and Lordan and Zabetakis [71]. 

#### 3.4.1. Fatty Acid Positional Distribution 

During digestion, fats undergo enzymatic hydrolysis by pancreatic lipases, which cleave FAs preferentially from the *sn-1* and *sn-3* positions, i.e., the outer positions of the triglyceride molecule, leaving the central *sn-2* FA attached to the glycerol molecule. These *sn-2* monoglycerides are freely absorbed regardless of the remaining FA type. The rate of absorption of free FAs (FFAs) emanating from the *sn-1* and *sn-3* positions depends on FA chain length with longer chain FAs (C12:0 to C18:0) less readily absorbed than shorter chain FAs (C6:0 to C10:0). As such positional esterification of FAs is important in terms of nutrition and energy provision to the neonate. Human milk has an unusually high proportion of C16:0 at the *sn-2* position (60% or more), which facilitates absorption and digestion, and has led to recent interest in the provision of infant formulae (IF) with this triglyceride structure. 

Vegetable oils, traditionally used in IF, have much lower levels of palmitic acid at *sn-2*. Comparison of the *sn-2* content of the milk fats examined in this study (*n* = 1; Table 7) demonstrate that cow cream had the highest level of palmitic acid (41.5%) with sheep the lowest at 29.5%. Others have reported similar values [72,73] for cow milk fat with rather less data available for other species [67,74,75]. Blasi et al. [76] reported % *sn-2* values for C16:0 of 44.1, 35.7, 27.2 and 40.9 for cow, goat, sheep and buffalo milk, respectively. Although comparatively lower than human milk with respect to *sn-2* C16:0 levels cow milk still represents a valuable source and is a promising substrate for lipase-induced structural modification of milk fat triglycerides for inclusion into IF products. In the case of *sn-1* and *sn-3* FA positioning, the results of the present study are similar to those published by Blasi et al. [76] and summarised by MacGibbon and Taylor [62] and Mohan et al. [61]. Studies suggest that the regiospecific binding of FA is likely to be genetically based [77]. The influence of the positional distribution of FAs on both nutrition and functional properties (melting and crystallisation) of lipid triglycerides remains poorly understood. 

#### 3.4.2. Phospholipids

The polar lipids of milk are the main constituents of the MFGM; the tri-layer film that stabilises milk fat globules against coalescence. The composition of the MFGM ranges widely, with polar lipids accounting for 30–75% and proteins making up 25–75% [78]. Polar lipids make up only 0.4–1% of total milk lipids. Phospholipids (PL) are a sub-class of polar lipids comprised of glycerophospholipids and sphingolipids, which have well-established nutraceutical properties [79]. The main polar lipids of milk are phosphatidylcholine (PC), phosphatidylethanolamine (PE), phosphatidylinositol (PI), phosphatidylserine (PS) and sphingomyelin (SM). In the present study, goat milk fat had the highest total % weight of PL, with buffalo the lowest (*n* = 1; Table 7). The average fat globule size is generally lower in small ruminant species, compared to cow (goat < sheep < cow), and has a greater proportion of PL (as seen in the content of goat milk fat), a requirement for the stabilisation of relatively higher fat globule surface areas. In contrast, buffalo cream has a higher fat content than the other species examined but also has larger fat globules, and accordingly has a lower proportion of polar lipids relative to total fat [69]. PE accounted for the highest proportion of PLs, with the exception of cow cream in which PC was marginally higher (31.4 vs. 28.6% of total PL). PI and PS values were the lowest overall. It should be noted that studies indicate that variability in the proportions of PLs in the MFGM can stem from the methods used for PL extraction rather than inter-species variation [80].

### 3.5. Confocal Microscopy

Confocal laser scanning in combination with fluorescently labelled probes was used to observe the fat structures from cow, sheep, goat and buffalo cream samples (*n* = 1). As illustrated in Figure 5 Row 1, Nile Red staining of lipids and Fast Green of protein showed protein located in the cream matrix and at the surface interface or the lipids. Nile Red has the ability to label the hydrophobic triacylglycerol molecules, which constitute the bulk of lipid material and exist within the core of the fat globule [21]. Cow and sheep cream showed occasional thick patches of proteins at the surface. Gallier et al. [20] reported these to be possible cytoplasmic crescents, which are considerably less evident in cow milk compared to human milk [81]. Buffalo cream showed larger areas of aggregated protein between fat droplets and appeared to have a larger globule size compared to the other species. Although limited variation existed amongst all samples. Goat cream showed considerable amounts of surface adsorbed proteins. 

Rh-PE staining can offer information on the lateral organisation of polar lipids located within the MFGM [21]. All creams stained positively for Rh-PE fluorescence indicating the presence of phospholipid layers around fat droplets (Row 2). In all species the distribution was heterogeneous and areas of richer domains can be observed, which has been previously reported as a liquid disordered phase, rich in unsaturated glycerophospholipids [20,21,82]. The fluorescently labelled WGA binds to sugar moieties of glycoproteins and glycolipids and so is used to detect the presence of oligosaccharide-containing molecules in the MFGM. Again, in all species the distribution was heterogeneous, and areas of richer domains can be observed (Row 3). Row 4 showed the dual staining of the Rh-PE and WGA.

## 4. Conclusions

In this study, the similarities and distinct differences in the MFG structure, protein and lipid composition of cow, buffalo, goat and sheep cream were reported. Although bulk mature milk representing the totality of commercial dairy herds was used to produce cream samples for investigation, it is important to recognize the need for a follow-up study that investigates the impact of feed, lactation stage and other factors on milk composition. Overall, the nutritional and functional value of all cream types was demonstrated. Furthermore, comparative analyses have revealed that buffalo, goat and sheep cream could have the potential to replicate the functional outcomes associated with the consumption of the cow milk fat globule membrane, including the support of the development of the infant immune and gastrointestinal system, cognitive function, protection from infection, and cardiovascular and muscular health. Further studies are warranted to confirm all health-promoting properties of the creams and to attribute these benefits to the MFGM. 

## Figures and Tables

**Figure 1 foods-10-02643-f001:**
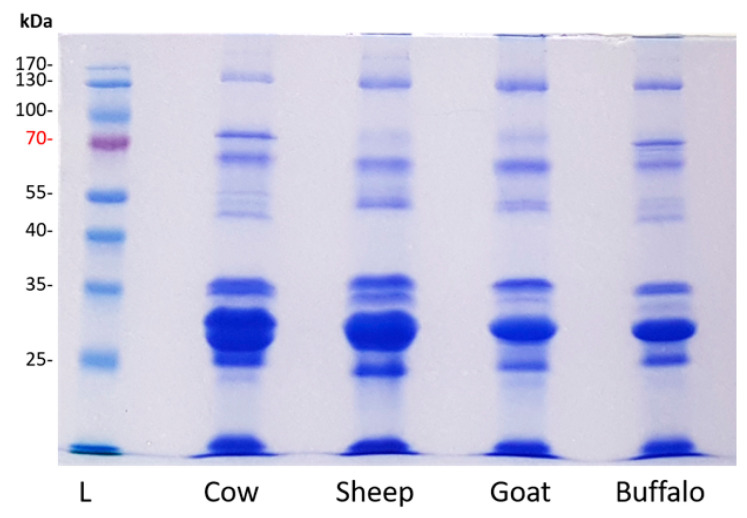
SDS-PAGE analysis of proteins isolated from a fraction of cream from cow, sheep, goat, and buffalo. L: prestained protein ladder with molecular weights indicated.

**Figure 2 foods-10-02643-f002:**
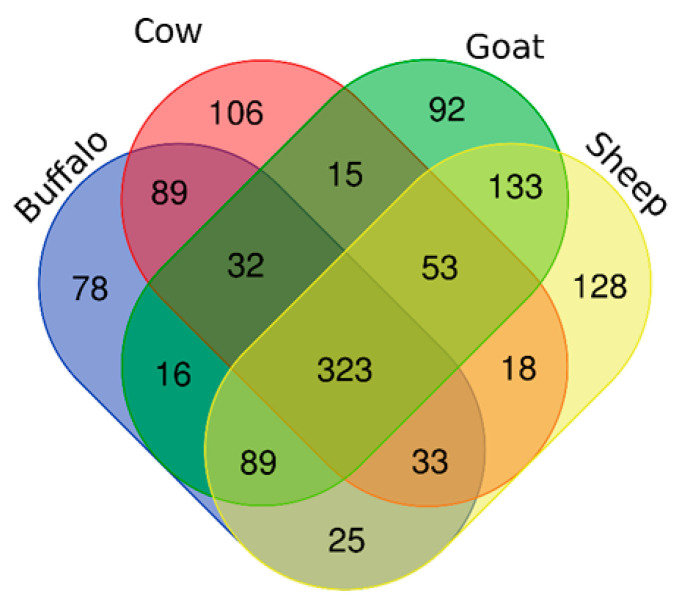
Venn diagram with the distribution of protein groups detected from buffalo, cow, sheep and goat cream samples. Image constructed using the Draw Venn Diagram tool at Bioinformatics & Evolutionary Genomics. Available online: http://bioinformatics.psb.ugent.be/webtools/Venn/ (accessed on 24 April 2020).

**Figure 3 foods-10-02643-f003:**
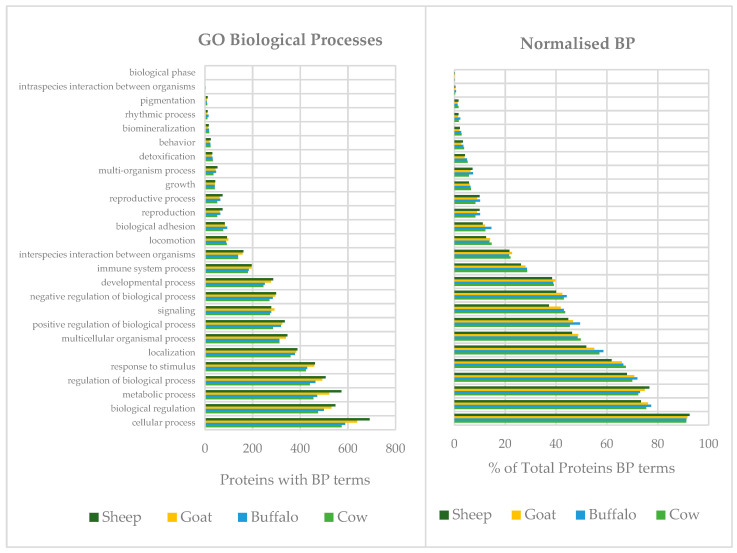
Summary of Biological Process (BP) terms of protein groups from analysis of cream from four species, with absolute numbers (**left**) and normalised to the total number of annotated proteins (**right**).

**Figure 4 foods-10-02643-f004:**
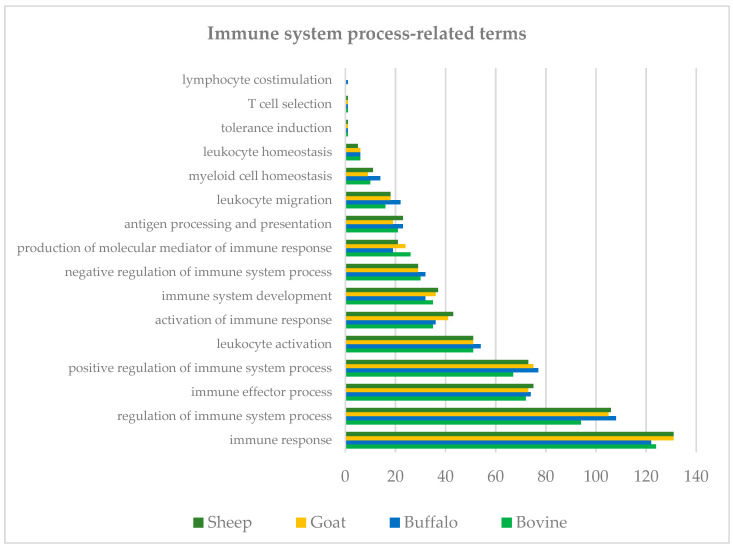
Summary of Biological Process immune system-related terms.

**Figure 5 foods-10-02643-f005:**
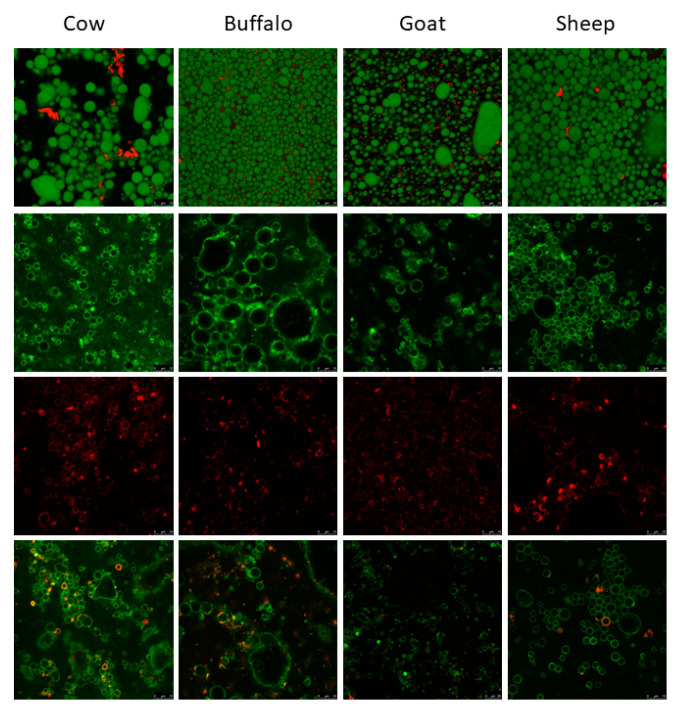
**Row 1:** Confocal micrographs showing milk fat globules stained with Nile Red (Green) and Fast Green FCF (red). The triacylglycerol core of the fat droplets is stained with Nile Red (green spheres) and the red indicates interfacial proteins. **Row 2:** Emission fluorescence of Rh-PE dye (green). The surface of fat droplets stained with Rh-PE indicating phospholipid coatings in the samples. **Row 3.** Emission fluorescence of WGA Alexa Fluor 488. Glycosylated molecules such as sugar residues from glycoproteins and glycolipids are shown by staining with WGA (red). **Row 4.** Confocal micrographs showing fluorescence emission of both the fluorescently labelled phospholipid Rh-PE (green) and lectin wheat germ agglutinin (red). Scale for all images = 10 µm. Column A = Cow cream, Column B = Buffalo cream, Column = Goat cream and Column D = Sheep cream.

**Table 1 foods-10-02643-t001:** Percentage of proteins in cream samples from cow, buffalo goat and sheep samples.

Species	Cow	Sheep	Buffalo	Goat
% Protein	5.47	6.03	3.68	5.12

**Table 2 foods-10-02643-t002:** Proteins identified in cow, buffalo, goat and sheep cream in the biological process GO terms “Antigen process and presentation”.

Antigen processing and presentation	**Accession Number**	**Protein Names**	**Cow**	**Buffalo**	**Goat**	**Sheep**
	I1VE56	RAS oncogene protein	✓	✓	✓	✓
	W5QAY5	Ras-related protein Rab-10	✓	✓	✓	✓
	W5QBQ8	Ras-related protein Rab-5C	✓	✓	✓	✓
	C0IZ95	RAB27A (RAB27A, member RAS oncogene family)	✓		✓	✓
	W5PDR1	RAB6A, member RAS oncogene family	✓	✓	✓	✓
	F1MNI4	RAB5B, member RAS oncogene family	✓	✓	✓	✓
	A0A452FMD4	Ras-related protein Rab-13	✓		✓	✓
	W5Q2D9	Ras-related protein Rab-5A-RAB5A	✓	✓	✓	✓
	A0A452G9U0	RAB35, member RAS oncogene family	✓	✓	✓	✓
	A0A452EMT1	RAB3D, member RAS oncogene family	✓	✓	✓	✓
	A6QR46	Ras-related protein Rab-6B			✓	
	W5P4F5	RAB27B, member RAS oncogene family				✓
	XP_006058006.1	Ras-related protein Rab-3A (Fragment)		✓	✓	
	XP_006080000.1	Ras-related protein Rab-8B		✓	✓	
	XP_006072022.1	Ras-related protein Rab-5A (Fragment)		✓		
	Q6QAT4	Beta-2-microglobulin	✓	✓	✓	✓
	P01888	Beta-2-microglobulin (Lactollin)	✓			
	A0A452DYV8	Alpha-2-glycoprotein 1, zinc-binding	✓	✓	✓	✓
	Q3ZCH5	Zinc-alpha-2-glycoprotein (Zn-alpha-2-GP) (Zn-alpha-2-glycoprotein)	✓	✓		
	XP_006053677.1	Alpha-2-glycoprotein 1, zinc-binding		✓		
	A0A452F1W3	Thrombospondin 1	✓	✓	✓	✓
	G5E513	Immunoglobulin heavy constant mu	✓	✓	✓	✓
	W5NXW9	Immunoglobulin heavy constant mu		✓		✓
	G5E5T5	Immunoglobulin heavy constant mu	✓	✓		
	A0A452EPC6	Immunoglobulin heavy constant mu			✓	
	A0A452DRV9	Prosaposin	✓			✓
	A0A3Q1MAG5	Proteasome activator complex subunit 1		✓		✓
	W5PXC6	Ig-like domain-containing protein				✓
	W5PI17	Ig-like domain-containing protein				✓
	XP_006052357.2	Ig-like domain-containing protein	✓	✓		✓
	XP_025116417.1	Ig-like domain-containing protein	✓	✓		
	XP_006041616.1	Calreticulin	✓	✓	✓	✓
	W5Q8S5	Aminopeptidase (EC 3.4.11.-)				✓

**Table 3 foods-10-02643-t003:** Proteins identified in cow, buffalo, goat and sheep cream in the biological process GO terms “Activation of immune response”.

Activation of immune response	**Accession Number**	**Protein Names**	**Cow**	**Buffalo**	**Goat**	**Sheep**
	Q3T169	40S ribosomal protein S3 (EC 4.2.99.18)	✓	✓	✓	✓
	E1BMJ0	Serpin family G member 1	✓	✓	✓	✓
	A0A452F9F6	Serpin family G member 1			✓	✓
	XP_006080796.1	SERPIN domain-containing protein		✓		
	F1MVS9	Mannan binding lectin serine peptidase 1	✓	✓	✓	✓
	XP_025142489.1	Mannan binding lectin serine peptidase 1		✓		
	Q7SIH1	Alpha-2-macroglobulin (Alpha-2-M)	✓		✓	✓
	A0A452EU27	Alpha-2-macroglobulin			✓	✓
	XP_025138969.1	Alpha-2-macroglobulin	✓	✓	✓	✓
	W5PIG2	Tyrosine-protein kinase (EC 2.7.10.2)	✓	✓		✓
	F1N261	Tyrosine-protein kinase (EC 2.7.10.2)	✓	✓	✓	✓
	A5PKG9	Tyrosine-protein kinase (EC 2.7.10.2)	✓	✓		
	F1MVK1	Anaphylatoxin-like domain-containing protein	✓			✓
	XP_025123384.1	Anaphylatoxin-like domain-containing protein		✓	✓	✓
	A0A452FKE5	Anaphylatoxin-like domain-containing protein		✓	✓	✓
	A0A452FTS0	Anaphylatoxin-like domain-containing protein			✓	
	XP_025123371.1	Anaphylatoxin-like domain-containing protein	✓	✓	✓	✓
	G5E513	Immunoglobulin heavy constant mu	✓	✓	✓	✓
	G5E5T5	Immunoglobulin heavy constant mu	✓	✓		
	W5NXW9	Immunoglobulin heavy constant mu		✓		✓
	A0A452EPC6	Immunoglobulin heavy constant mu			✓	
	A0A3Q1M3L6	Ig gamma-3 chain C region (Fragment)	✓	✓	✓	✓
	W5NPT7	Butyrophilin subfamily 1 member A1	✓	✓	✓	✓
	P18892	Butyrophilin subfamily 1 member A1 (BT)	✓			
	A3EY52	Butyrophilin subfamily 1 member A1	✓	✓	✓	✓
	F1MZQ4	Butyrophilin subfamily 1 member A1	✓	✓	✓	✓
	A0A452ELE7	Complement C2			✓	
	Q2UVX4	Complement C3	✓	✓	✓	✓
	A0A452DXE2	Complement C3			✓	✓
	XP_006045164.2	Complement C3		✓	✓	✓
	A0A452EW11	Complement C7			✓	
	F1N045	Complement component C7	✓	✓	✓	✓
	XP_025144420.1	Complement C8 beta chain		✓		
	A0A452ENB6	Complement C9			✓	✓
	XP_006063582.2	Complement C9	✓	✓		
	Q3T0A3	Complement factor D (EC 3.4.21.46) (Adipsin) (C3 convertase activator) (Properdin factor D)	✓			✓
	A0A452FFD7	Complement factor H		✓		✓
	W5PDS4	Complement C1q A chain			✓	✓
	W5PDP6	Complement C1q B chain				✓
	A0A452FR95	Complement factor B	✓	✓	✓	✓
	A0A3Q1LRP5	Complement factor B	✓	✓		
	A5YBU9	Complement factor B	✓			
	W5PH95	Ig-like domain-containing protein	✓	✓		✓
	A0A452EE69	Ig-like domain-containing protein			✓	
	W5Q7I2	Ig-like domain-containing protein			✓	✓
	W5QFH6	KRAS proto-oncogene, GTPase	✓		✓	✓
	W5PD71	Pentaxin (Pentraxin)		✓	✓	✓
	P23907	Major prion protein (PrP) (CD antigen CD230)		✓	✓	✓
	A0A452F2C6	Ezrin	✓	✓	✓	✓
	P31976	Ezrin (Cytovillin) (Villin-2) (p81)		✓		
	W5NXJ3	TED_complement domain-containing protein			✓	✓
	A0A452E5L3	Protein tyrosine phosphatase receptor type J			✓	
	XP_025121828.1	Peptidylprolyl isomerase	✓	✓	✓	✓
	W5NRI1	A2M_N_2 domain-containing protein			✓	✓
	W5NU00	A2M domain-containing protein				✓
	W5PGT9	Ig epsilon chain C region (Fragment)				✓
	G3N342	Ig epsilon chain C region (Fragment)	✓			
	W5P5T4	NTR domain-containing protein				✓
	XP_006064867.1	NRAS proto-oncogene, GTPase	✓	✓	✓	✓
	XP_006057277.2	Toll-like receptor 4		✓		
	W5NQC1	Presenilin (EC 3.4.23.-)	✓	✓		
	P50448	Factor XIIa inhibitor (XIIaINH)	✓			
	A0A3Q1LPG0	Uncharacterized protein	✓		✓	
	A0A3Q1N3I9	Uncharacterized protein	✓			
	A0A452F0Q6	Uncharacterized protein	✓		✓	✓
	A0A452F0Q1	Uncharacterized protein			✓	

**Table 4 foods-10-02643-t004:** Proteins identified in cow, buffalo, goat and sheep cream in the biological process GO terms “Production of molecular mediator of immune response”.

Production of molecular mediator of immune response	**Accession Number**	**Protein Names**	**Cow**	**Buffalo**	**Goat**	**Sheep**
	W5QHZ5	Ig kappa chain C region			✓	✓
	F1MZ96	Ig kappa chain C region	✓			
	A0A452F4S4	Ig-like domain-containing protein	✓	✓	✓	
	F1MLW8	Ig-like domain-containing protein	✓			✓
	F1N160	Ig-like domain-containing protein	✓		✓	
	W5PSQ7	Ig-like domain-containing protein	✓	✓	✓	✓
	A0A3Q1LWV8	Ig-like domain-containing protein	✓	✓		✓
	A0A3Q1MSF6	Ig-like domain-containing protein	✓	✓	✓	✓
	A0A452EVZ5	Ig-like domain-containing protein			✓	
	A0A452E8D3	Ig-like domain-containing protein	✓	✓	✓	✓
	A0A3Q1NI92	Semaphorin 7A	✓			
	XP_006046407.2	Semaphorin 7A (John Milton Hagen blood group)		✓		
	A0A452G0Z2	Semaphorin 7A (John Milton Hagen blood group)			✓	✓
	W5QAB1	Hemopexin			✓	✓
	Q3SZV7	Hemopexin	✓	✓	✓	✓
	XP_006042170.1	Hemopexin		✓	✓	
	P26201	Platelet glycoprotein 4 (Glycoprotein IIIb) (GPIIIB) (PAS IV) (PAS-4) (Platelet glycoprotein IV) (GPIV) (CD antigen CD36)	✓	✓		
	A0A452G1U2	CD36 molecule	✓	✓	✓	✓
	W5Q6I2	CD36 molecule		✓		
	W5Q6N3	CD36 molecule	✓		✓	✓
	Q0GC71	Toll-like receptor 2 (EC 3.2.2.6) (CD antigen CD282)	✓	✓	✓	✓
	Q95LA9	Toll-like receptor 2 (EC 3.2.2.6) (CD antigen CD282)	✓	✓	✓	✓
	A0A452G4X3	Toll-like receptor 2			✓	✓
	W5Q0A3	Toll-like receptor 2				✓
	F1N720	Toll-like receptor 2	✓			
	XP_006057277.2	Toll-like receptor 4		✓		
	A0A452FI14	Apolipoprotein A1	✓	✓	✓	✓
	XP_025122634.1	Apolipoprotein A1	✓	✓		
	A0A3Q1MAA6	Growth arrest specific 6	✓			
	P21214	Transforming growth factor beta-2 proprotein (Milk growth factor) (MGF) [Cleaved into: Latency-associated peptide (LAP); Transforming growth factor beta-2 (TGF-beta-2)]	✓		✓	✓
	XP_006050765.1	DnaJ heat shock protein family (Hsp40) member B9	✓	✓	✓	✓
	G3MZ88	J domain-containing protein	✓			
	P81644	Apolipoprotein A-II (Apo-AII) (ApoA-II) (Antimicrobial peptide BAMP-1) (Apolipoprotein A2) [Cleaved into: Proapolipoprotein A-II (ProapoA-II); Truncated apolipoprotein A-II (Apolipoprotein A-II(1-76))]	✓	✓	✓	
	Q6QAT4	Beta-2-microglobulin	✓	✓	✓	✓
	P01888	Beta-2-microglobulin (Lactollin)	✓			
	Q1ZZU7	Macrophage migration inhibitory factor (MIF) (EC 5.3.2.1) (L-dopachrome isomerase) (L-dopachrome tautomerase) (EC 5.3.3.12) (Phenylpyruvate tautomerase)	✓		✓	✓
	W5PAG0	Lysine--tRNA ligase (EC 6.1.1.6) (Lysyl-tRNA synthetase)			✓	✓
	A0A452FWT3	Sphingosine kinase 2			✓	

**Table 5 foods-10-02643-t005:** Fat content (g/100g) in cream samples from cow, buffalo, goat and sheep cream samples.

	Cow (g/100g)	Sheep (g/100g)	Goat (g/100g)	Buffalo (g/100g)
Fat Content	84.4	87.4	86.6	91.2

**Table 6 foods-10-02643-t006:** Fatty acid composition in the *sn-2* position of the triglyceride molecules in cream samples from cow, buffalo, goat and sheep cream samples.

Fatty Acids	Triglycerides				Répartition %							
					Sn2				Sn1+3			
	Cow	Sheep	Goat	Buffalo	Cow	Sheep	Goat	Buffalo	Cow	Sheep	Goat	Buffalo
C4:0	3.5	3.6	3	4.6	2.6	2.7	2.6	3	97.4	97.3	97.4	97
C6:0	2	2.7	2.5	2.5	2.8	4.6	3.7	4.5	97.2	95.4	96.3	95.5
C8:0	1.1	2.7	2.6	1.4	17.4	23.2	21	28.4	82.6	76.8	79	75.2
C10:0	2.6	7.9	8.9	2.8	21.6	26.4	25.3	27.9	78.4	73.6	74.7	72.1
C12:0	3	4.3	4.2	3.3	37.9	37.3	42.3	40.1	62.1	62.7	57.7	59.9
C13:0	0.2	0.2	0.2	0.1	37.2	36.2	38.6	41.6	62.8	63.8	61.4	58.4
C14:0	11.2	9.1	10.2	12.4	51.6	43.6	55	47.1	48.4	56.4	45	52.9
C14:1	0.9	0.1	0.2	0.8	37.6	46.1	43.6	45.1	62.4	53.9	56.4	54.9
C15:0	2.4	2	1.6	1.9	46.3	42.8	46.7	43.8	53.7	57.2	53.3	56.2
C16:0	32.2	19.7	27.2	27.8	41.5	29.5	35.3	34.3	58.5	70.5	64.7	65.7
C16:1	1.7	1.6	1.1	2.1	51.5	42.8	49.6	42.8	48.5	57.2	50.4	51.8
C17:0	1.8	1.5	1.6	1.1	33.1	33.8	33	38.4	66.9	66.2	67	61.6
C17:1	0.3	0.2	0.3	0.2	55.1	55.3	58.1	58.8	44.9	44.7	41.9	41.2
C18:0	10.1	9.9	8.9	9.7	16.7	29	22.3	24.8	83.3	71	77.7	75.2
C18:1	22.6	26.6	23.1	23.6	27	39.2	32.7	34.8	73	60.8	67.3	65.2
C18:2	2.9	6.2	3.6	4.4	31.3	42.9	39.9	34.9	68.7	57.1	60.1	65.1
C18:3 (n-3)	1	1.1	0.4	0.9	24.9	29.6	32.2	27.5	75.1	70.4	67.8	72.5
C20:0	0.2	0.1	0.2	0.1	-	63.8	53.6	19.7	-	36.2	46.4	80.3
C20:1	0.1	<0.05	<0.05	0.1	-	-	-	-	-	-	-	-
C20:3 (n-6)	<0.05	<0.05	<0.05	<0.05	-	-	-	-	-	-	-	-
C20:4 (n-6)	0.1	0.1	0.2	<0.05	25.7	49.1	29.5	-	74.3	50.9	70.5	-
C20:5 (n-3)	0.1	0.1		0.1	20.8	44.2		-	79.2	55.8		-
C22:0	0.1	0.1	<0.05	0.1	-	83.9	-	-	-	16.1	-	-
C22:1	<0.05	<0.05	<0.05	<0.05	-	-	-	-	-	-	-	-
C22:5 (n-3)		0.2	0.1	0.1		89.2	-	52.1		10.8	-	47.9
C22:6 (n-3)	<0.05	0.1	<0.05	<0.05	-	90.6	-	-	-	9.4	-	-
C24:0	<0.05	<0.05	<0.05	<0.05	-	-	-	-	-	-	-	-
C24:1	<0.05	<0.05	<0.05	<0.05	-	-	-	-	-	-	-	-

**Table 7 foods-10-02643-t007:** Phospholipids content in cream samples from cow, buffalo, goat and sheep samples. Values are expressed as weight percentages. PC—Phosphatidylcholine, LPC—Lyso-phosphatidylcholine, PI—Phosphatidylinositol, LPI—Pyso-phosphatidylinositol, PS—Phosphatidylserine, LPS—Lyso-phosphatidylserine, PE—Phosphatidylethanolaime, SPH—Sphingomyelin, APE—N-Acyl-phosphatidyletholamine, LPE—Lyso- phosphatidyletholamine, DPG—Diphosphatidylglycerol, PG—Phosphatidylglycerol, PA—Phosphatidic Acid, LPA—Lyso-phosphatidic Acid.

Phospholipids	Cow (Weight %)	Sheep (Weight %)	Goat (Weight %)	Buffalo (Weight %)
PC	0.11	0.1	0.12	0.09
1-LPC	-	-	-	-
2-LPC	-	-	-	-
PI	0.02	0.01	0.01	-
LPI	-	-	-	-
PS-Na	0.03	0.02	0.04	0.01
LPS	-	-	-	-
SPH	0.09	0.09	0.12	0.07
PE	0.1	0.12	0.13	0.1
LPE	-	-	-	-
APE	-	-	-	-
PG	-	-	-	-
DPG	-	-	-	-
PA	-	-	-	-
LPA	-	-	-	-
Other PL	-	-	-	-
Sum	0.35	0.34	0.42	0.27

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
