# Peer review of "Comparative Structural and Compositional Analyses of Cow, Buffalo, Goat and Sheep Cream"

_foods, 2021, doi:10.3390/foods10112643_

Round 1
Reviewer 1 Report
The manuscript provide sufficient knowledge and include all relevant references.
The experiment was carried out correctly, it is a pity that the production of cream from cow's milk took place in a different industrial manner. The research methodology should be supplemented with details and references to already approved and published methods.
Congratulations on carrying out such comprehensive proteomics and lipid studies. Review of the literature and discussion of the results introduces a lot of new information to the general knowledge.
Author Response
Dear Editor and Reviewers,
Thank you for giving us the opportunity to submit a revised draft of the manuscript entitled “Comparative structural and compositional analyses of cow, buffalo, goat and sheep cream”. We appreciate the time and effort that you have dedicate to providing valuable feedback on our manuscript. We have carefully revised the manuscript according to your comments (highlighted in red)(see attached) and provide a point-by-point response to reviewers’ comments and concerns. We are grateful for the insightful comments that add valuable improvements to our manuscript.

Reviewer 2 Report
I consider this manuscript very interesting.
However, I have some suggestions to improve the text:
- Lines 35 and 37: I think there should be dots there to indicate the end of sentences.
- In my opinion the Abstract should not contain abbreviations of scientific terms. The last sentence in the Abstract must match the title of the manuscript - milk fat or cream?
- Line 75: Please add reference number "Hodgkinson and colleagues".
- Lines 93-95: Please replace the term "alternative cream sources" with "cow, buffalo, goat and sheep cream". The word "alternative" may suggest the use of plant substitutes for a dairy cream.
- “Materials and Methods” section: Please provide the total number of analyzed samples of cream obtained from each animal species and the number of replications of the analyzes performed.
- Lines 100-104: Please add centrifugation parameters: eg speed (or centrifugal force value) and temperature. To what fat content was the cream centrifuged? Was the centrifugation a one-step or a two-step process?
- Lines 104-105: Please specify how long the storage was at 4oC.
- Lines 118-120: Please describe briefly the principle of the methods used in ITERG.
- Line 124: Where did cream powder come from? How to compare the condition of proteins in cream powder with proteins from fresh cream?
- Lines 184-186: The protein content of milk varies greatly and depends on many factors, including the animal's breed, health, lactation period, number of lactations and feed. I suggest that you take this into account when discussing the results.
- Lines 189-191: In my opinion, it is difficult to discuss the protein content of the creamer without discussing the fat content, the more so as it was a cream obtained by centrifugal separation rather than spontaneous stand-by separation. How do obtained results relate to the protein content of commercial creams with standardized fat content? There is no discussion of the results with the literature data here. The manuscript does not fully identify the origin of the samples (e.g. including animal species, breed, and feed). Is the received data consistent with the data of other authors?
- Line 190: What is the "cream fraction" all about? Wasn't that an analysis of the whole cream?
- Table 1: no data on the number of repetitions and standard deviation (or SE value). Were the differences statistically significant? Please complete it.
- “Proteomic analyses of cream samples from four species” section: There is no discussion of the results with the literature data here. The protein composition varies greatly and depends on many factors. I suggest that you take this into account when discussing the results.
- Figures 3 and 4: no data on the number of repetitions and standard deviation (or SE value). Were the differences statistically significant? Please complete it.
- Line 386: “seasonality”, well, in this respect also the samples of creams used in this study should be characterized.
- Tables 5 and 6: no data on the number of repetitions and standard deviation (or SE value). Were the differences statistically significant? Please complete it.
- Lines 508-511: Could the observed differences result from different methods and parameters of separating cream from milk? Or maybe the reasons were the parameters of storage and transport of samples from the moment of milk separation to the time of analysis?
Author Response

(The authors gave the same response as above.)

Reviewer 3 Report
Dear enclosed some remarks from your research paper. This study seems original by its approach. However, I have some concerns concerning some points of the paper. I hope that those comments will permit to improve you paper.
My principal concern is about the analysis of membrane of milk fat globule. The authors never point out in the paper if the molecules they analyzed are really extracted from the fat globule membrane. If this is not clear enough, the reader can ask the question how the authors can be sure that the molecules presented are not simply present in the different phase of the milk cream.
Moreover, in the materials and methods part two kind of centrifugations were used to separate the cream from the skimmed milk. This can have an effect on the characteristics and composition of cream extracted. This should be discussed in the text.
All the manuscript need to be formatted in the foods journal format. Please correct.
Line 32. Does not make sense. Please correct.
Line 34. Does not make sense.
Line 36. Give meaning of Abbreviations!
Line 37. Add a space between protein and Functional.
Line 41. Space between “species.” and “Palmitic”.
Line 42: delete the point.
General comments on the abstract. The results presented in the abstract are not precise enough. Moreover, the conclusion of the abstract is not in accordance with the results previously presented. How those results can be associated to the "potential application in infant formula".
Line 51 please give reference for this sentence.
Line 54. What the authors means by "all those factors". No factors are presented before in the text.
Line 55. For the FAOSTAT data, the authors must give the year associated to the percentage presented.
Line 75: please give a reference number for Hodgkinson and colleagues.
Line 103. Is this difference in separation system can have an effect on fat composition/cream characteristics? Please precise/or comment this point.
Please precise also the breeds of animals.
Materials and methods. The authors must precise how the samples were transported from farms to laboratory and how the samples were transferred from France to Ireland, if this is the case.
Materials and methods. The authors should precise the characteristics of the centrifuge systems (rate, number of plates...).
Materials and methods. More precision should be given concerning the number of repetition performed for each sample. This must be specified for all the analysis. More details needs to be given concerning the experiments.
Line 380. Please correct the sentence.
Please correct table 5. They is an error in the structuration of the table.
Correct also the names of the fatty acids. This is not the classical nomenclature. C18:1 and not C18:01 !
Author Response

(The authors gave the same response as above.)

Round 2
Reviewer 2 Report
I can see that the Authors tried to answer all the questions and comments of the reviewers and they managed to do so. I understand that it was not possible to revise the manuscript to such an extent as to take into account all the suggestions of the reviewers. From my point of view the manuscript has been sufficiently corrected and supplemented. I have no further comments on this text.